# Ring Opening Polymerization of Six- and Eight-Membered Racemic Cyclic Esters for Biodegradable Materials

**DOI:** 10.3390/ijms25031647

**Published:** 2024-01-29

**Authors:** Andrea Grillo, Yolanda Rusconi, Massimo Christian D’Alterio, Claudio De Rosa, Giovanni Talarico, Albert Poater

**Affiliations:** 1Department of Chemical Sciences, Università degli Studi di Napoli Federico II, Via Cintia, 80126 Napoli, Italy; and.grillo@studenti.unina.it (A.G.); yolanda.rusconi@unina.it (Y.R.); massimochristian.dalterio@unina.it (M.C.D.); claudio.derosa@unina.it (C.D.R.); talarico@unina.it (G.T.); 2Scuola Superiore Meridionale, Largo San Marcellino 10, 80138 Napoli, Italy; 3Institut de Química Computacional i Catàlisi and Departament de Química, Universitat de Girona, c/ Maria Aurèlia Capmany 69, 17003 Girona, Spain

**Keywords:** ring-opening polymerization, polylactic acid, poly(lactic-*co*-glycolic acid), poly(3-hydroxybutyrate), Salen, stereoselectivity

## Abstract

The low percentage of recyclability of the polymeric materials obtained by olefin transition metal (TM) polymerization catalysis has increased the interest in their substitution with more eco-friendly materials with reliable physical and mechanical properties. Among the variety of known biodegradable polymers, linear aliphatic polyesters produced by ring-opening polymerization (ROP) of cyclic esters occupy a prominent position. The polymer properties are highly dependent on the macromolecule microstructure, and the control of stereoselectivity is necessary for providing materials with precise and finely tuned properties. In this review, we aim to outline the main synthetic routes, the physical properties and also the applications of three commercially available biodegradable materials: Polylactic acid (PLA), Poly(Lactic-*co*-Glycolic Acid) (PLGA), and Poly(3-hydroxybutyrate) (P3HB), all of three easily accessible via ROP. In this framework, understanding the origin of enantioselectivity and the factors that determine it is then crucial for the development of materials with suitable thermal and mechanical properties.

## 1. Introduction

Due to the global population increase, the demand for goods and raw materials is growing exponentially [1]. Among the immediate consequences of this phenomenon are the depletion of available resources, the significant environmental impact resulting from increased greenhouse gas emissions [2,3], and subsequent global warming [4]. The development of increasingly eco-sustainable production processes that reconcile the needs of the population with environmental preservation is, therefore, becoming the focus of numerous research endeavors [5]. The role that Green Chemistry plays in today’s society is crucial. It represents a conception of chemistry focused on the development of new materials and processes with reduced environmental impact for sustainable development. Sustainable development is understood as “development that meets the needs of the present without compromising the ability of future generations to meet their own needs” [6]. Therefore, adopting the principles underlying sustainable chemistry does not mean limiting industrial activity but ensuring an increase in industrial productivity by reducing all types of costs [7,8], with environmental impact being the foremost concern. To achieve this goal in the shortest possible time, it is advisable to choose alternative reagents, seek more efficient technologies, and prioritize safer transformations [9]. All of this should be conducted following specific guidelines aimed at:Improving energy processes through recycling and the use of renewable sources.Optimizing waste management cycles.Substituting hazardous products with safer alternatives.Following the principles of atom economy.Minimizing the use of auxiliary substances (solvents, separating agents).Developing new catalysts.Reducing the E-factor.Increasing plant safety.

Without this approach, the chemical industry cannot be considered sustainable [10]. It is evident from above that sustainability is based on three fundamental components: economic, social, and environmental, and it can only be achieved through close collaboration between researchers, producers, and consumers [11]. This short review project falls within this field, focusing on the study, particularly through Density Functional Theory (DFT) calculations thanks to robust results with stereoselective *rac*-Lactide and *rac*-Diolide and regioselective 3-methyl glycolide polymerization [12]. In addition, it discusses the synthesis of industrially relevant polymeric materials from renewable sources using a mechanism of ring-opening polymerization (ROP) promoted by chiral systems. In detail, the review aims to improve the knowledge of stereoselectivity in polymer catalysis.

## 2. Classification of Polymeric Materials

It is possible to classify polymeric materials based on the eco-friendliness of raw materials and the ease of recycling or degradation. This classification (schematically summarized in Figure 1) allows for the assessment of the life cycle of polymers, considering both the environmental impact of the raw materials used in their production and the available options for managing polymer waste.

The raw materials used to produce polymers can be either fossil-based or renewable. Polymers derived from raw materials like petroleum could have a significant environmental impact due to their non-renewable origin and related gas emissions during their production. On the other hand, polymers obtained from biomass or agricultural waste offer a more sustainable option as they reduce dependence on non-renewable resources and can contribute to a reduction in gas emissions [13].

The ease of recycling or degradation of polymeric materials is a crucial aspect of the reduction of environmental impact. Polymers like polyethylene terephthalate (PET) are widely recyclable and can be transformed into new products. Other polymers, like expanded polystyrene (EPS), are more challenging to recycle and often end up in landfills. However, new technologies and processes are continuously developing to improve the recycling of less recyclable polymers, promoting a circular polymer economy. On the other hand, polymer degradation can be a solution to reduce the accumulation of plastic waste in the environment. Some polymers degrade more easily through biological or chemical processes, reducing their long-term impact on the ecosystem. However, careful consideration of degradation methods is essential to avoid potential negative impacts such as the release of toxic substances or soil and water pollution [14].

For all these reasons, the classification of polymers based on the eco-friendliness of raw materials and the ease of recycling or degradation plays a significant role in evaluating their environmental impact. The use of renewable raw materials and the promotion of efficient recycling and degradation processes can contribute to reducing the environmental impact of polymers and promoting sustainable waste management. Poly(lactic acid) (PLA) and Poly(3-hydroxybutyrate) (P3HB) are examples of polymers that fall within this classification, offering more eco-friendly alternatives to traditional plastics. However, considering the specific environmental conditions and implementing appropriate waste management practices are essential to maximize their ecological benefits. Nevertheless, PLA and P3HB are biodegradable polymers that have gained significant attention due to their various applications across different industries.

## 3. Poly(Lactic Acid)

PLA is a biodegradable polymer (see Figure 2), primarily derived from renewable sources like corn starch or other sugars. It has been widely adopted as a substitute for traditional plastics in applications such as packaging, containers, and disposable utensils. PLA is considered eco-friendly because its raw materials are renewable, and it can be naturally degraded by bacteria or enzymes, reducing plastic waste accumulation. However, the degradation of PLA requires specific environmental conditions, such as optimal temperature and humidity, which may not be readily available in all landfills or natural environments. In particular, due to the chiral nature of lactic acid, lactides can exist in three distinct isomers: D-LA (*R*,*R*), L-LA (*S*,*S*), and *meso*-LA. Stereoselectivity is a critical factor in the ring-opening polymerization of cyclic monomers, particularly for synthesizing polymers like PLA.

### 3.1. Physical Properties and Applications

PLA has a unique combination of physical properties that make it suitable for various applications [15,16,17]. Certainly, PLA is derived from renewable resources, possesses biodegradability, and showcases compelling physical and mechanical characteristics that rival those of conventional olefin-based polymers [18,19]. In addition, it is worth noting that the physical properties of PLA can be further modified by blending it with other polymers or incorporating additives. By altering the composition, PLA can exhibit enhanced mechanical properties, improved thermal stability, or additional functionalities, depending on the specific requirements of the application [20]. It is now clear that understanding PLA’s physical characteristics is essential for the optimization of its use in different industries. First, PLA exhibits good thermal stability, with a melting temperature (*T*_m_) typically ranging from 150 to 180 °C [21]. PLA is also characterized by its optical properties. It has excellent transparency, allowing the production of clear and visually appealing products. For all these reasons, PLA is widely used in the production of sustainable packaging materials. Studies by Chandra and Rustgi [22] and Siracusa et al. [23] have explored PLA’s potential as a packaging material, highlighting also its excellent barrier properties, thermal stability, and biodegradability. Anderson and Shive investigated PLA as a suitable material for food packaging [24], emphasizing its ability to extend the shelf life of products while ensuring their safety and reducing environmental impact. The biocompatibility of PLA has led to its use in biomedical and pharmaceutical applications. Studies by Langer and coworkers [25] and Grijpma and Pennings [26] highlighted the potential of PLA in tissue engineering, drug delivery systems, and biodegradable implants.

Another notable property of PLA is its relatively low glass transition temperature (*T*_g_), which is around 60–65 °C. This means that PLA transitions from a rigid, glassy state to a softer, rubbery state at relatively low temperatures. The low *T*_g_ enables PLA to be easily molded and shaped during processing, making it compatible with various manufacturing techniques, including injection molding and 3D printing. Liu et al. investigated PLA’s use as a 3D printing filament [27], noting its ease of printing, dimensional accuracy, and potential for producing intricate structures.

PLA has also been explored as a sustainable alternative in the textile industry. Studies by Abdelrazek et al. demonstrated the feasibility of producing PLA-based fibers [28], highlighting their biodegradability, comfort, and potential for applications in clothing and textiles.

Finally, PLA films have been utilized in agriculture as biodegradable mulch films. Taib et al. [21] examined the use of PLA films for weed suppression, moisture retention, and soil temperature regulation, offering sustainable alternatives to traditional plastic mulches.

### 3.2. Possible Strategies for PLA Synthesis

PLA can be synthesized through various routes, offering distinct advantages in terms of efficiency and control over polymer properties (see Figure 3). First, ROP is the most widely used method for PLA synthesis, typically employing lactide monomers as starting materials. Particularly, depending on the stereochemistry of the initial monomer, various microstructures of PLA can be obtained, ranging from isotactic PDLA and PLLA to isotactic stereoblock PLA, as well as heterotactic and syndiotactic PLA. The first two microstructures can be derived from enantiopure D- and L-LA monomers or, in principle, their racemic mixture (*rac*-LA) through stereoselective polymerization [29,30]. Industrially, since L-LA is a cheap and commercial feedstock, there is no need to employ the *rac*-LA mixture to synthesize PLLA (Figure 3A) [31]. Indeed, L-lactic acid (and not D-lactic acid) is specifically synthesized by bacteria through biomass fermentation. The isotactic stereoblock PLA (Figure 3B) is particularly noteworthy due to the stereocomplexation of enantiomeric blocks [2], resulting in a higher *T*_m_ (up to 30–50 °C) compared to enantiopure PLLA and PDLA. The achievement of this microstructure is primarily accomplished through either (1) the sequential addition of enantiopure L-LA and D-LA, utilizing a fast and living catalyst [32], or (2) employing a stereoselective catalyst starting from *rac*-LA as the reaction feed. The latter method, with its cost-effective and straightforward reaction feed, would be preferable but, as we will see later, poses a non-trivial chemical challenge. The syndiotactic microstructure, leading to a semicrystalline polymer [33], can exclusively be synthesized from *meso*-LA using stereoselective catalysis [34]. Finally, the heterotactic microstructure is achievable both from the stereoselective ROP of *rac*-LA and *meso*-LA but does not find any practical application due to its inability to crystallize (Figure 3C).

The stereoselective ROP of LA, which can be achieved using proper catalysts, will be thoroughly discussed in the following sections [35]. Direct polycondensation could also involve the reaction of lactic acid monomers to form PLA, with the removal of water as a byproduct. Kim and Woo developed an efficient direct polycondensation method for high molecular weight PLA using a catalyst system consisting of SnCl_2_·2H_2_O [36]. Unfortunately, the polycondensation stops when the equilibrium is reached. To proceed with the reaction, it is necessary to remove the water from the reaction system.

In addition, enzymatic polymerization offers a green approach to PLA synthesis. Studies by Kobayashi et al. explored the use of lipases as catalysts for the enzymatic polymerization of lactide [37]. They demonstrated high efficiency and control over polymer properties, including molecular weight and stereochemistry.

Neat polymerization involves the heating and the melting of lactide and the subsequent polymerization employing a robust catalyst, for example, Sn(Oct)_2_. This process is actually employed in the industry mainly for the production of PLLA, using L-LA produced via biomass fermentation. Indeed, the absence of hazardous solvents such as dichloromethane and toluene makes this process green and technologically feasible. For this kind of process, recent research has been pushing toward the findings of new catalytic systems based on metals less hazardous than Sn, as for example, Zn [38]. Shinno et al. [39] investigated the solid-state polymerization of L-LA and PLLA using two different solvent-free approaches. Both of them led to an only remaining monomer ratio exceeding 5 wt % and allowed an enhancement of the molecular mass of the PLLA, involving the reactive -OH terminus of the polymeric chain.

The last method consists of a reactive extrusion that combines polymerization and melt processing, offering a continuous and efficient synthesis route. Jacobsen et al. investigated the reactive extrusion of lactide using a corotating, closely intermeshing twin-screw extruder [40]. They demonstrated efficient polymerization and control over PLA properties by adjusting reaction conditions, such as temperature and monomer feed rate.

## 4. Poly(Lactic-*co*-Glycolic Acid)

PLGA, a copolymer of lactic and glycolic acid, is a biocompatible polyester obtained from renewable raw materials such as corn starch and sugar cane (Figure 4). The FDA approved this biodegradable polymer for biomedical applications, mainly for tissue scaffolds or drug delivery vehicles in the form of micro and nanoparticles [41,42]. Specifically, this material constitutes the polymer matrix to which an active pharmaceutical ingredient (API) can be added in order to deliver the drug to specific parts of the body.

### 4.1. Physical Properties and Applications

The property that makes PLGA interesting is the biodegradation through hydrolysis of its ester linkages and following bioassimilation through the inclusion of the hydrolyzed monomers in the Krebs cycle. Considering its applications in drug delivery, one of the main features to be controlled is the degradation rate of the polymer, which depends on several external factors (pH, temperature, etc.) as well as tacticity, molecular weight, monomers ratio, and monomer order [43,44].

Polymer composition is a key factor since it determines the hydrophilicity and, therefore, the degradation. The weight loss of the polymer increases with the percentage of glycolic acid since the GA portions are more hydrophilic [44,45]. Samples with a 50:50 ratio of LA/GA and an alternation of the two LA and GA monomeric units throughout the polymer display linear (and tunable) degradation rates. Random PLGA, instead, displays first a fast degradation followed by slow degradation rates and, in general, uncontrolled drug release kinetics [46,47].

### 4.2. Possible Strategies for PLGA Synthesis

As already introduced for PLA synthesis, there are several methods to synthesize PLGA. One is the direct polycondensation of lactic and glycolic acids, the main drawback being the difficult removal of water to shift the equilibrium and the consequent low molecular weights. A step-growth segmer condensation has also been proposed by a few groups, in which the repeated sequence depends on the starting oligomer employed [48,49]. However, this method does not result in proper molecular weight control.

Industrially, the most widely used method is the ring-opening polymerization of lactide and glycolide, using tin (II) bis (2-ethylhexanoate) (Sn(Oct)_2_) as catalyst. However, the toxicity associated with tin compounds limits its use for biomedical applications [50].

Ring-opening polymerization can also be employed with the cyclic cross dimer of lactic and glycolic acid, called 3-methyl glycolide (3-MeG). 3-MeG displays two attack sites: the LA acyl site and the GA acyl site. Regioselective attack at only one of the two sites results in perfectly alternating PLGA (Figure 5). In 2000, Dong et al. carried out the homopolymerization of MeG using tin octanoate [51]. Organocatalytic systems, such as phosphazene base/alcohol systems, have been proposed for this reaction, solving the toxicity issue associated with tin [52]. Recently, Lu et al. proposed a methodology for the synthesis of alternating PLGA using chiral aluminum complexes already tested in the isoselective and syndioselective ROP of *rac*-LA and *meso*-LA [53,54,55].

## 5. Poly(3-hydroxybutyrate)

Poly(3-hydroxybutyrate) (P3HB) is another biodegradable polymer that is produced by bacteria through fermentation processes (see Figure 6). It is a natural polymer and can be derived from renewable sources such as vegetable oils or organic waste. P3HB possesses interesting physical and mechanical properties, making it suitable for various applications, including packaging and biomedical devices. It is biodegradable in marine and terrestrial environments due to the action of microorganisms present in the environment, contributing to waste reduction.

### 5.1. Physical Properties and Applications

P3HB exhibits good mechanical properties. It has a relatively high tensile strength, allowing it to withstand stretching forces without breaking. Hoffmann et al. [56] measured the tensile strength of pure P3HB, noticing that this value at 180 °C was between 32 Mpa and 28 MPa in the first 100 s of stretching, reaching about 18 Mpa after 600 s. Young’s modulus, on the other hand, does not change a lot during the first 600 s at 180 °C, assuming values between 2.2 and 2.3 GPa. These properties make P3HB suitable for applications requiring durable and robust materials, such as packaging, consumer goods, and even engineering components.

P3HB also possesses good thermal stability. It is stable at temperatures typically encountered in everyday use and during processing. For these reasons, one of the key applications of P3HB is in the packaging field [57]. Two of the related notable physical properties of P3HB are *T*_g_ and *T*_m_ [56,58]. The first is typically around 4 °C, while the second is commonly around 180 °C. This high *T*_m_ enables P3HB to retain its structural integrity even at high temperatures. It ensures dimensional stability and allows for various processing techniques, including injection molding and extrusion. Another important physical property of P3HB is its biocompatibility. It is a biologically derived polymer (and, for this reason, suitable for medical applications) that is non-toxic and compatible with living tissues. P3HB is also characterized by its biodegradability, meaning it can be naturally broken down by microorganisms into harmless byproducts. It is a renewable and environmentally friendly alternative to petroleum-based plastics. Its biodegradability allows for the reduction of the environmental impact and the potential to close the loop in the lifecycle of products, particularly in applications where disposability is necessary. In terms of optical properties [59], P3HB is usually transparent or translucent. It enables the production of visually appealing products that require clarity, such as packaging films or containers. However, the transparency of P3HB can be modified by adding pigments or other additives to achieve desired colors or opacity. Furthermore, P3HB has a low water absorption [60], which makes it resistant to moisture.

### 5.2. Possible Strategies for P3HB Synthesis

P3HB synthesis can be achieved through diverse pathways, each offering unique advantages in terms of efficiency and control over polymer characteristics.

Currently, microbial fermentation stands as the primary industrial method for P3HB production (Figure 7). This underscores why researchers have explored different combinations of carbon sources, varying ratios, and bacterial species types.

Drawing from experimental data, scientists have employed various strategies to boost intracellular accumulation and polymer yield of PHAs. These approaches include screening strains, refining cultivation, and operational procedures, and optimizing carbon and nitrogen sources within the culture medium. In addition to the carbon source and the microbic species, process conditions are crucial during the production of P3HB as well. Indeed, pH, temperature, equipment, RPM (rotation per minute), time, and aeration are also fundamental parameters to monitor to increase the yield.

Certain bacteria can produce P3HB even during the cell growth phase [61]. For these strains, a continuous feed system is necessary to achieve higher yields.

Unfortunately, despite the efficiency of biological approaches in the P3HB synthesis, they are not able to provide enough quantity to satisfy the market demand. This is the reason why organometallic catalysis may be the most efficient way to maximize production, guaranteeing the quality of the final polymer.

The initial attempts at metalorganic catalysis for the synthesis of P3HB were conducted starting from β-butyrolactone (β-BL). Carpentier et al. [62], through yttrium-based catalysis (Figure 8), attempted to obtain syndiotactic P3HB starting from β-butyrolactone.

Bruckmoser et al. [63], on the other hand, tried to develop another Y-based catalytic system (Figure 9) in order to stereoselectively catalyze the ROP of β-butyrolactone.

Figure 10 resumes the possible ways to organometallically catalyze the ROP of β-BL. As shown, both the syndiotactic and the isotactic P3HB can be synthesized.

However, the ROP of β-butyrolactone, carried out through metal-organic catalysis, proves to be disadvantageous mainly for two reasons: firstly, a relatively high activation energy is required to initiate a four-carbon atom cycle. Additionally, to obtain the final isotactic P3HB from a racemic mixture, it is necessary to develop complex systems capable of promoting the coordination of one monomer over the other with opposite chirality. For this reason, in order to achieve P3HB with a high degree of isotacticity, reducing the activation energy barrier, Tang and Chen [64,65] conducted the polymerization of a diolide (8DL, cyclic dimer of 3-hydroxybutyric acid) to obtain P3HB through ROP mechanism promoted by different Salen catalysts [66] (Figure 11).

By using a suitable initiator, dichloromethane, as a solvent and controlled reaction conditions, they successfully achieved P3HB in good yield and desired molecular weight. Displaying two stereogenic centers, the advantage of starting from the eight-membered diolides is the possibility of obtaining different stereoregular materials by ROP of the possible diastereoisomers: (*R*,*R*)-8DL, (*S*,*S*)-8DL and *meso*-8DL (Figure 12).

Currently, industrial production of P3HB does not typically involve the use of Salen catalysts. This is because bacterial process offers better stereoregularity in the final material affecting the thermal and mechanical properties of P3HB.

However, as reported by Xie et al. [67], researchers are currently studying new catalysts in order to improve stereoselectivity and conversion during the ROP mechanism of *rac*-diolides for the P3HB synthesis. It was claimed that Salen catalysts could be the best solution to substitute the bacterial process. Studying the reaction mechanisms and the different factors that influence stereoselectivity is therefore crucial to understanding how to build a competitive industrial process.

## 6. Importance of Stereo- and Regioselectivity in Cyclic Monomers ROP

Stereo- and regioselectivity play a crucial role in the ring-opening polymerization (ROP) of cyclic monomers for the synthesis of polymers like P3HB, PLA, and PLGA. The chiral nature of cyclic monomers, such as lactic acid for PLA, γ butyrolactone for P3HB, and methyl glycolide for PLGA, contributes to the formation of polymers with specific stereoisomeric configurations and sequences. The ability to control stereoselectivity is essential in achieving polymers with the desired properties [68,69,70].

In the synthesis of P3HB, stereoselectivity determines the isotactic configuration of the polymer, which influences its thermal and mechanical properties. Stereoselective copolymerization was employed by Tang et al. [71], incorporating flexible ε-caprolactone and γ-butyrolactone units into P3HB to overcome the high brittleness while taking advantage of poly(3-hydroxybutyrate) high crystallinity and crystallization rate.

Regarding PLA, the stereoselectivity influences the tacticity of the polymer, which in turn determines its crystallinity and mechanical and rheological properties [72,73]. As aforementioned, the isotactic stereoblock polylactic acid (PLA), mainly achievable from the stereoselective ROP of *rac*-LA, stands out notably because of the stereocomplexation between enantiomeric blocks [2]. Moreover, Chile et al. have highlighted the significance of controlling stereoselectivity in the polymerization of lactic acid to obtain isotactic PLA with the highest values of intrinsic viscosities and hydrodynamic radii (as functions of molecular weight, *M*_w_) for *iso*-PLAs [74], followed by *hetero* and then *syndio*-PLAs. The glass transition temperature (*T*_g_) of PLA is also influenced by its isotacticity. Several scientific studies have investigated the relationship between isotacticity and the *T*_g_ of PLA. For example, in a study by Urayama et al. [75], it was found that increasing the isotactic content of PLA led to an increase in its *T*_g_. The researchers prepared PLA samples with different L-content and D-content levels and measured their *T*_g_ using differential scanning calorimetry (DSC). They observed that PLA with higher isotactic content exhibited higher *T*_g_ values, indicating a correlation between isotacticity and *T*_g_. Lastly, for PLGA, regioselective ring-opening occurring at only one of the two possible attack systems of MeG allows complete sequence control, leading to the formation of a perfectly alternating material whose degradation rates are suitable for drug delivery applications.

## 7. Salen Catalysts for Cyclic Monomers ROP Catalysis

### 7.1. Advantages of Salen Catalysts in Cyclic Monomers ROP Catalysis

Over the past two decades, a multitude of homogeneous metal-based catalysts have been synthesized with the aim of achieving stereocontrolled ring-opening polymerization (ROP) of racemic lactide (*rac-*LA). Among these catalysts, Salen-based catalysts (illustrated in Figure 13) have gained global recognition as a platform offering numerous advantages in the control of stereoselectivity during the ROP of lactides. The success of Salen-based catalysts can be attributed primarily to two factors: (a) their straightforward and cost-effective synthesis, which is also reflected in facile tuning due to the availability of a broad range of commercially accessible, variously substituted precursors; (b) their exceptional compatibility with diverse metal centers. This compatibility arises from the structural composition of these ligands, which consist of two salicylaldimine moieties connected by a bridge, thereby establishing a distinctive chiral environment surrounding the metal center.

In principle, they were synthesized in 1889 by Combers, who prepared the first Salen ligand [66] and its related Cu complex. Since then, thanks to their general easy synthesis process (Figure 14) described by Horminirium et al. [76], Salen systems and their correspondent metal complexes have been synthesized and used in stereoselective catalysis studies [77,78,79].

In general, Salen catalysts offer several advantages in controlling the stereoselectivity during lactides and diolides ROP, leading to the formation of polymers with specific stereoisomeric configurations. Their most important characteristics are the relative steric and electronic properties. By modifying the substituents on the salicylaldimine moieties or the bridge, the steric hindrance and electronic environment around the metal center can be adjusted. This tunability enables the control of the catalyst’s reactivity and selectivity, leading to precise control over the formation of stereoisomeric configurations in the resulting polymers [80,81].

Salen catalysts have already proven to be effective in controlling the stereoselectivity during ROP. For instance, Carpentier and coworkers highlighted important advances in yttrium-based stereoselective ROP of cyclic esters toward the synthesis of some stereoregular polyhydroxyacids [82], and they have been able to synthesize polymers with syndiotactic and isotactic microstructures just changing the substituents on the catalyst ligand.

Regarding the ROP of *rac*-LA, actually, the first reported stereoselective Salen catalyst was the Spassky system (Figure 15) [83], which showed a high isoselectivity, represented by the optical purity (OP = ${\left[\alpha \right]}_{D}^{25}$/156) in Table 1. Computational DFT studies were executed to investigate the reaction mechanism and the origin of stereo- and regioselectivity of *rac*-LA [84], *meso*-LA [9] and MeG [55] ROP catalyzed by Spassky’s system focusing on the two transition states (TSs) and two intermediates as shown in Figure 16.

### 7.2. Metal Centre Influence on Catalyst Stereoselectivity

Salen catalysts have great compatibility with various metal centers. Different transition metals, such as titanium, tin, and aluminum, can be incorporated into Salen ligands to modulate the stereoselectivity of lactide and diolide ROP. Each metal center imparts distinct reactivity and selectivity, allowing for the fine-tuning of the polymerization process and the resulting polymer properties. The choice of the central metal in Salen ligands then plays a crucial role in the stereoselectivity during the ROP of lactides and diolides.

Recently, Rusconi et al. reported a computational study regarding the origin of stereoselectivity of Salen catalysts in the ROP of *rac*-LA, comparing four systems bearing the same ligand but different metals (Al, Sc, Y and La, Figure 17) [85]. The calculations led to the conclusion that increasing the size of the central metal relaxes the catalytic structure (detectable and predictable [86] through a %*V*_Bur_ analysis [87,88] and steric maps [89,90]), allowing access to both enantiomers (L-LA and D-LA), inevitably leading to a loss of stereoselectivity.

Similar features were observed by Bakewell et al. [91], who explored the stereoselective polymerization of lactide using similar La, Y, and Lu-based phosphasalen catalysts (Figure 18), demonstrating how the differences in the metal dimensions are crucial in determining the stereocontrol (Table 2) [92].

In particular, the isotacticity index (*P*_i_) was determined by analyzing the homonuclear decoupled NMR spectrum according to Coudane et al.’s methodology [93]. In addition, the correspondent order of reaction rates is inversely related to the metallic, covalent radius (La = 2.07 Å > Y = 1.90 Å > Lu = 1.87 Å [94]) that, therefore, affects both catalyst stereoselectivity and activity.

### 7.3. Ligands Influence on Catalyst Stereoselectivity

Salen ligands, like the metal center, play a critical role in determining the stereoselectivity during cyclic monomers ROP, as demonstrated by Tang, Chen and coworkers [64,65]. They showed the correspondent stereoselectivity of various ligand typologies (Figure 19).

More specifically, as shown in Table 3, the values that most explain the resulting polymer isotacticity are the probability of *meso* linkages between HB units (*P*_m_) and the concentration of isotactic triad made up of two adjacent meso diads (*mm*), determined by ^13^C and ^1^H NMR spectroscopy. According to the results, the best system between the ones they used for the stereoselective ROP of *rac*-DL is catalyst d.

### 7.4. Bridge Influence on Catalyst Stereoselectivity

The term “bridge” refers to the part of the Salen ligands that connects the two nitrogen atoms. By carefully selecting the bridge, it is possible to modulate the stereoselectivity and control the formation of specific stereoisomeric configurations in the resulting polymers. For example, studies by Gibson and coworkers [76] and Feijen and coworkers [95] demonstrate the variation of PLA isotacticity just varying the bridge of Al-based Salen catalysts (see Figure 20).

As reported in Table 4, Feijen’s catalyst is the most stereoselective. His preference is for the (*S*,*S*)-lactide and, as a consequence, for the synthesis of PLLA. This demonstrates the role of the bridge’s steric hindrance and chirality. Stereoselectivity is, for instance, favored by a chiral, flexible, and steric-hindered bridge.

### 7.5. Solvent and Initiator Influence on Catalyst Stereoselectivity

Solvents can influence the reaction kinetics, the coordination of the metal center, and the interaction between the catalyst and the monomer, thereby affecting the stereoselectivity and control over the formation of specific stereoisomeric configurations in the resulting polymers.

One example is the use of polar aprotic solvents such as dichloromethane or toluene. These solvents have been found to promote the stereoselective ROP of lactide, leading to the formation of isotactic polylactide (PLA). On the other hand, nonpolar solvents like hexane or cyclohexane can also influence stereoselectivity. These solvents typically lead to slower reaction rates and lower stereoselectivity due to reduced solubility and coordination of the catalyst. Polar coordinating solvents such as THF are directly involved in inferring stereoselectivity to the reaction. Indeed, heterotactically enriched PLA can be obtained even using achiral catalysts.

A study conducted by Chisholm et al. [96] explored the effects of different solvents and initiators on the stereoselectivity of a single ring-opening reaction catalyzed by Salen catalysts (Figure 21), showing that polar aprotic solvents favor the formation of isotactic PLA.

## 8. Future Perspectives and Challenges

The ROP of racemic cyclic esters for the production of biodegradable materials presents several challenges [97,98,99]:→Enantioselective Polymerization:
→Issue: Common ROP catalysts may not be enantioselective, leading to the formation of atactic or non-stereoregular polymers.→Challenge: Developing catalysts for the control of the chirality of the polymer chain is essential for producing materials with predictable and desirable properties.→Molecular Weight Distribution:
→Issue: The ROP of racemic cyclic esters may result in broad molecular weight distributions.→Challenge: Achieving a narrow molecular weight distribution is important for ensuring consistent material properties. This often involves optimizing reaction conditions and catalyst systems.→Reaction Kinetics:
→Issue: Racemic cyclic esters may exhibit different reaction kinetics compared to their pure stereoisomers.→Challenge: Understanding and controlling the reaction kinetics is essential for achieving the desired polymerization rates and preventing side reactions.→Polymerization Rate:
→Issue: Racemic cyclic esters may polymerize at different rates for each enantiomer.→Challenge: Balancing the polymerization rates of different enantiomers to obtain well-defined copolymers or blends can be challenging and may require careful tuning of reaction conditions.→Mechanical Properties:
→Issue: Polymers derived from racemic cyclic esters may exhibit variations in mechanical properties due to the lack of stereoselectivity.→Challenge: Optimizing reaction conditions and catalyst systems to achieve consistent mechanical properties in the resulting polymers is a significant challenge.→Biodegradability:
→Issue: The racemic nature of the polymer may influence its biodegradability.→Challenge: Ensuring that the resulting polymer maintains the desired biodegradability characteristics while addressing stereochemistry challenges is a complex task.→Catalyst Design:
→Issue: Many traditional catalysts may not be suitable for achieving stereochemistry control in the ROP of racemic cyclic esters.→Challenge: Designing catalysts that can effectively control stereochemistry while maintaining reactivity and selectivity is an ongoing area of research.

Addressing these challenges requires interdisciplinary efforts involving organic chemistry, polymer chemistry, and materials science. Researchers are continually exploring new catalysts, reaction conditions, and methodologies to overcome these hurdles and advance the development of biodegradable materials [100], derived from racemic cyclic esters.

## 9. Conclusions

Biodegradable polymeric materials, obtainable via the easily accessible ROP of cyclic diesters, represent valuable alternatives to most common polyolefins in terms of physical and mechanical properties and eco-friendliness. With the aim of greener alternatives in polymerization, achieving precise material properties relies on controlling stereo- and regioselectivity, emphasizing the importance of understanding the factors determining enantioselectivity. This knowledge is crucial for developing materials with tailored thermal and mechanical properties, promoting a sustainable and environmentally friendly approach to polymer synthesis.

Overall, this review aims to establish what has been conducted so far and, above all, to give the foundations of where the field can go. Surely, DFT calculations are and will continue to be the search engine in the coming times. Particularly, interesting insights were obtained by analyzing the combination of the metal role with suitable ligands to understand the stereoselectivity in the ROP of *rac*-LA [89]. In fact, by using a common chiral Salen ligand, the role of the metal center was revealed to be fundamental to explain the experimental preference of the (*R*,*R*)-1 system for *SS*-LA monomer in *rac*-LA ROP. However, further elements of complexity are due to the dynamic behavior in the solution (distinct wrapping modes around the metal center) and to the monomer enantiofaces (*RR*-LA and *SS*-LA select opposite monomer enantiofaces) in the RDS. Despite this complexity, the general trend that larger metals reduce steric hindrance (%*V*_Bur_), allowing access to both enantiomers and compromising stereoselectivity, can be sorted out. It is clear that aluminum (Al), although reporting higher stereoselectivities, may not be ideal for achieving high activities. Metals with higher atomic radii, such as Sc, Y, or La, reduce stereoselectivity, but they offer a promising scaffold for more active catalysts. Developing stereoselective Salen systems with metals other than Al may involve emphasizing steric hindrance on the Salen ligand or increasing monomer size. And what is missing? For example, future investigations should focus on the stereoselective polymerization to get applications [101], in particular of *rac*-LA by using a cheap and active catalyst based on Mg [102] and of *rac*-diolide, an eight-membered cyclic diester, using highly sterically encumbered Salen-Y [103] and Salen-La catalysts to unveil if this is the right direction to follow [103,104]. Actually, given the available results, to improve the knowledge of stereoselectivity [105], particularly in polymer catalysis [65,68,69,106], now is the right time to combine the existing data as it was big data for future machine learning efforts [86,88] to generate more active catalysts [107].

## Figures and Tables

**Figure 1 ijms-25-01647-f001:**
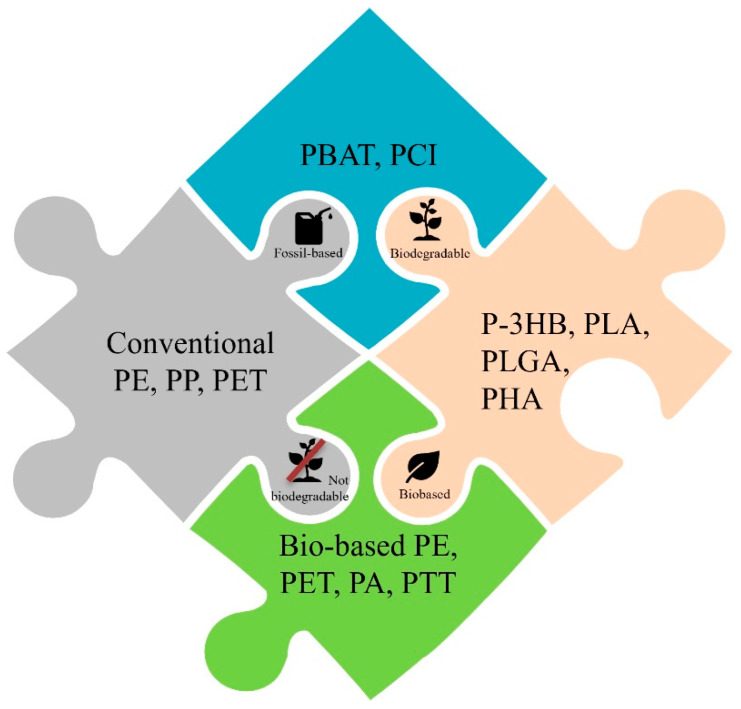
Schematic classification of polymeric materials based on the eco-friendliness of raw materials and the ease of degradation.

**Figure 2 ijms-25-01647-f002:**
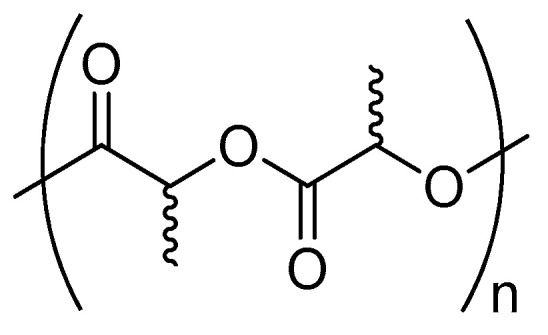
Polylactic acid (PLA).

**Figure 3 ijms-25-01647-f003:**
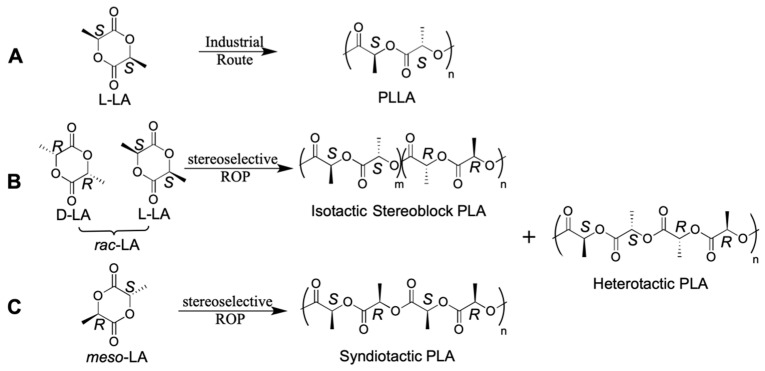
The ROP chemical equation of the catalyzed PLA synthesis from L-LA (**A**), *rac*-LA (**B**), and *meso*-LA (**C**).

**Figure 4 ijms-25-01647-f004:**
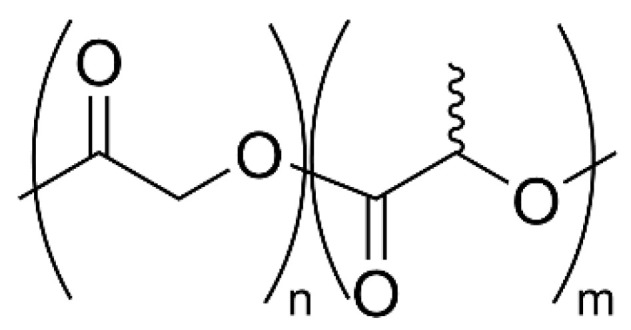
Poly(lactic-*co*-glycolic acid) (PLGA).

**Figure 5 ijms-25-01647-f005:**
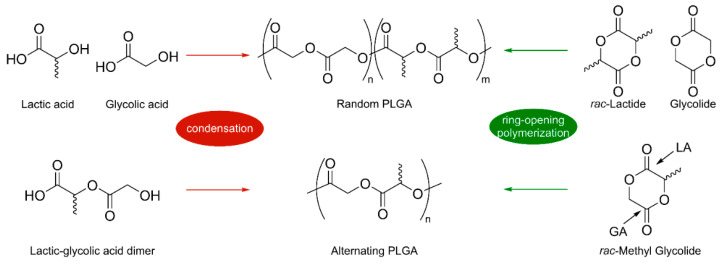
PLGA synthesis through polycondensation or ring-opening polymerization.

**Figure 6 ijms-25-01647-f006:**
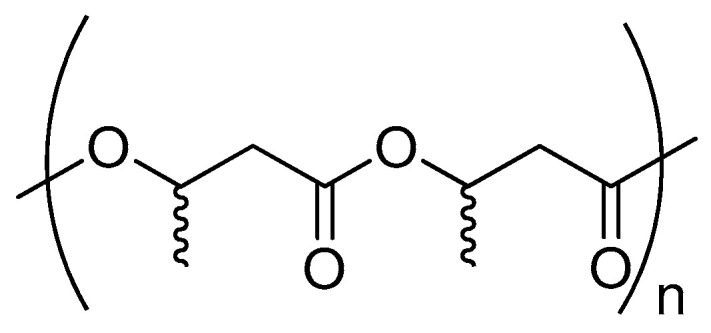
Poly(3-hydroxybutyrate) (P3HB).

**Figure 7 ijms-25-01647-f007:**
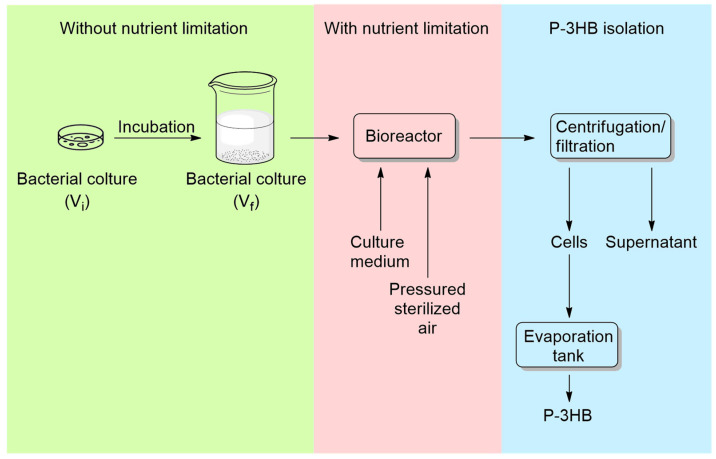
Schematization of the typical P3HB industrial production process.

**Figure 8 ijms-25-01647-f008:**
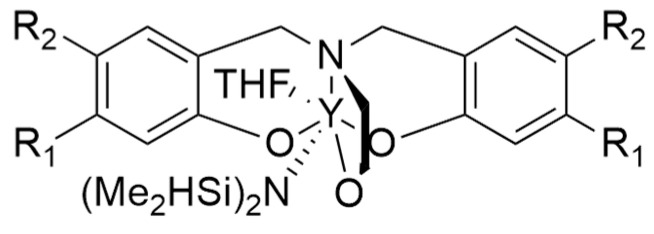
The Y-based catalysts used by Carpentier et al. [62] with R_1_ = R_2_ = CMe_3_; R_1_ = R_2_ = CPhMe_2_; R_1_ = CPh, R_2_ = Me.

**Figure 9 ijms-25-01647-f009:**
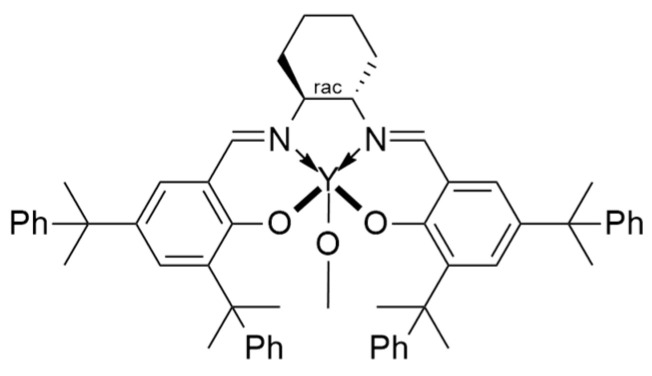
Y-based catalyst used by Bruckmoser et al. [63].

**Figure 10 ijms-25-01647-f010:**
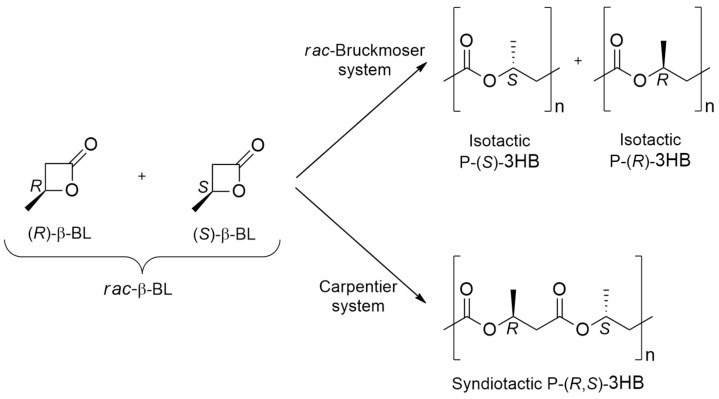
The ROP of β-butyrolactone leading to P3HB synthesis.

**Figure 11 ijms-25-01647-f011:**
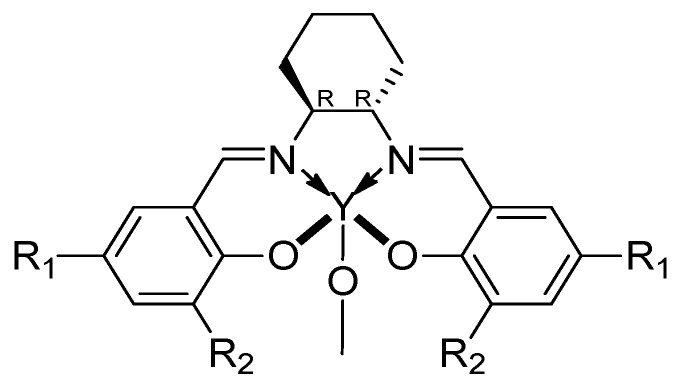
The Y-based catalysts used by Tang et al. with R_1_ = R_2_ = *t*Bu; R_1_ = *t*Bu and R_2_ = F; R_1_ = R_2_ = CMe_2_Ph; R_1_ = Me and R_2_ = CPh_3_ [64,65].

**Figure 12 ijms-25-01647-f012:**
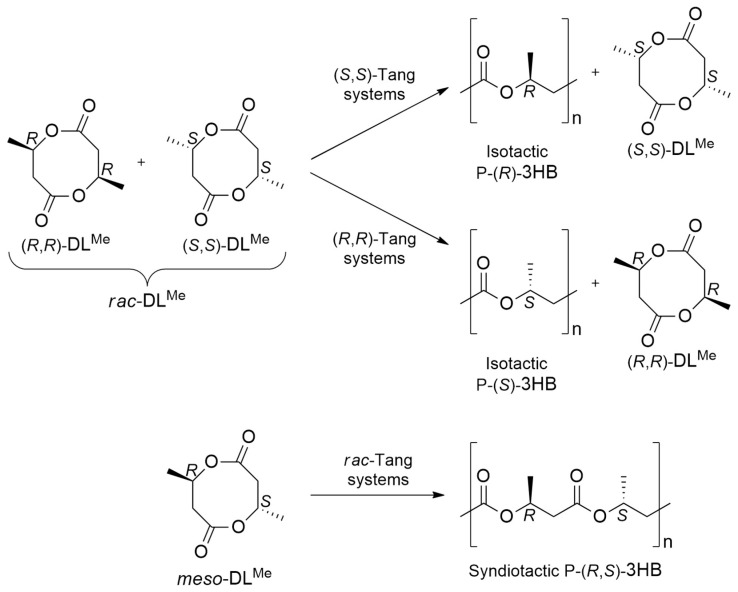
The ROP chemical equation of catalyzed P3HB synthesis by Tang et al.’s systems [64,65].

**Figure 13 ijms-25-01647-f013:**
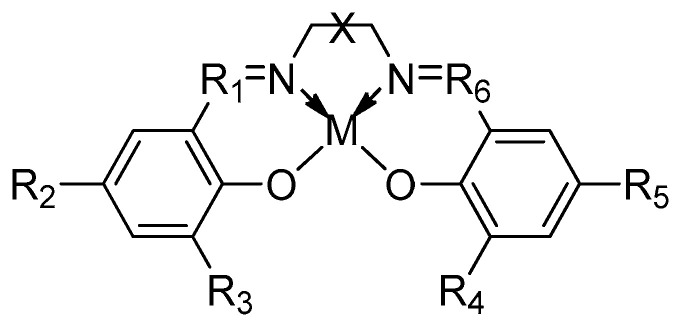
The Salen ligands coordinated to a general metal center “M”.

**Figure 14 ijms-25-01647-f014:**
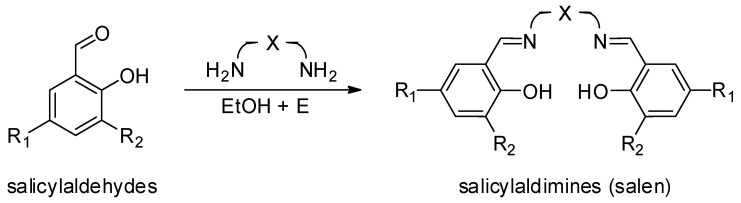
The Salen ligands synthesis process described by Gibson et al. [76].

**Figure 15 ijms-25-01647-f015:**
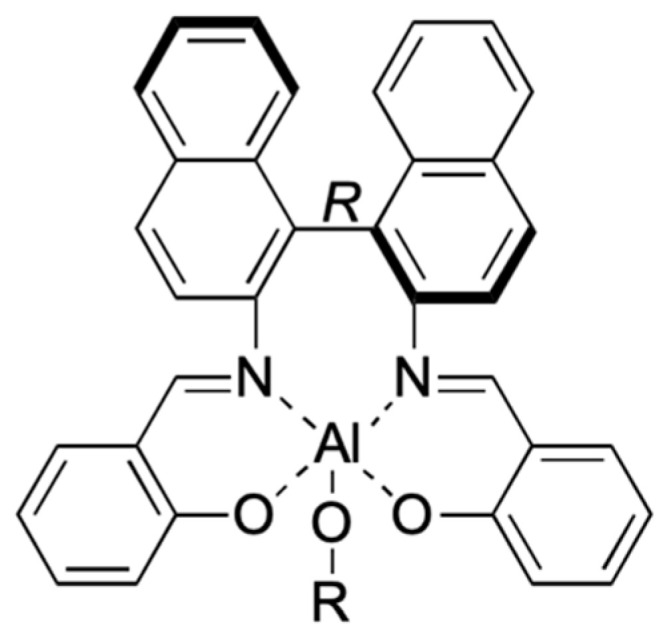
The salen-based catalyst used by Spassky et al. [83].

**Figure 16 ijms-25-01647-f016:**
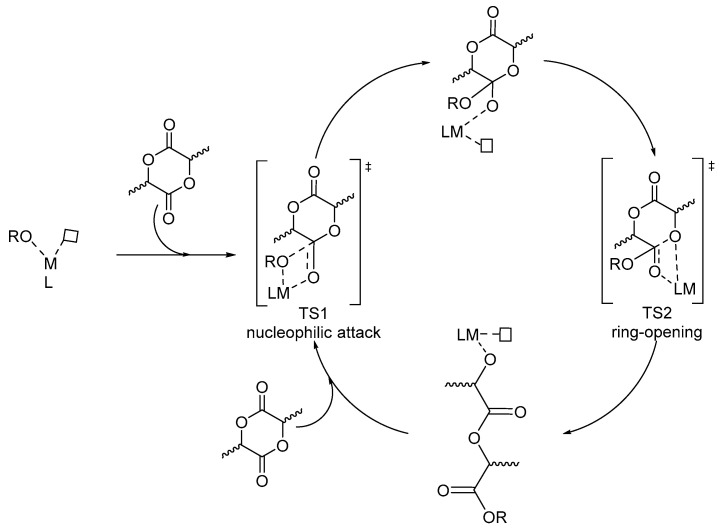
The metal-catalyzed ROP mechanism for *rac*-LA.

**Figure 17 ijms-25-01647-f017:**
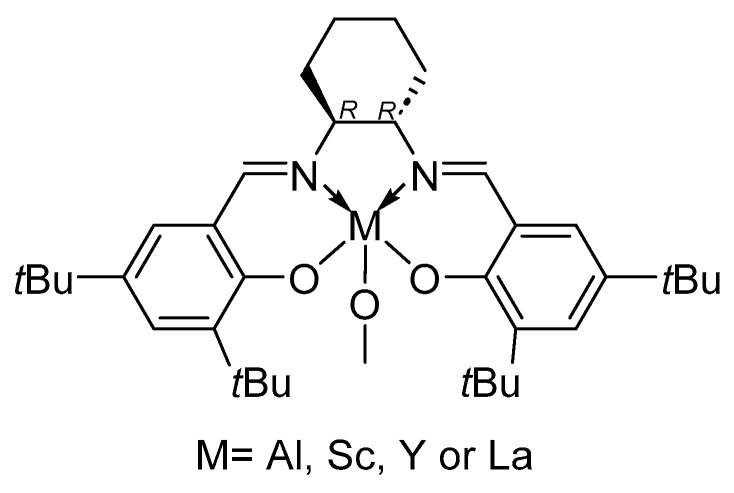
The salen-based catalyst studied by Rusconi et al. with R = *t*Bu and M = Al, Sc, Y, La [85].

**Figure 18 ijms-25-01647-f018:**
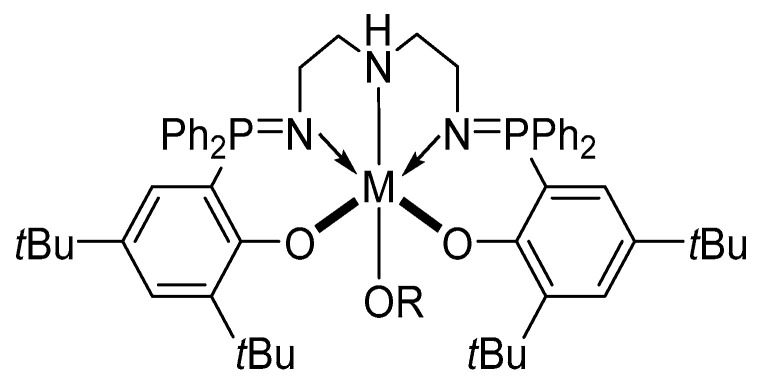
The phosphasalen-based catalyst used by Bakewell et al. with R = *^t^*Bu and M = Y, La, Lu [91].

**Figure 19 ijms-25-01647-f019:**
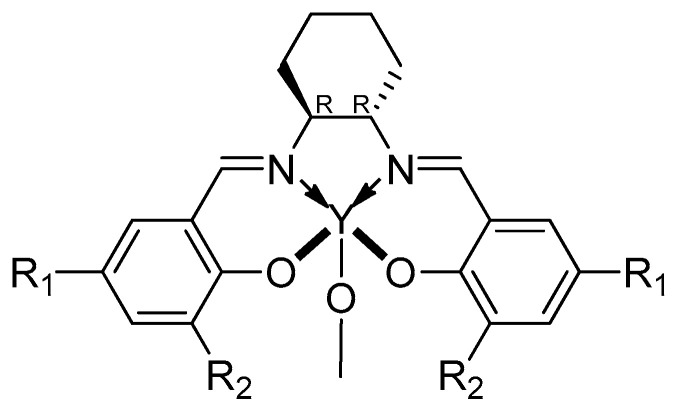
The Y-based catalysts used by Tang, Chen and coworkers [64,65] with R_1_ = R_2_ = *t*Bu; R_1_ = *t*Bu and R_2_ = F; R_1_ = R_2_ = CMe_2_Ph; R_1_ = CPh_3_ and R_2_ = *t*Bu.

**Figure 20 ijms-25-01647-f020:**
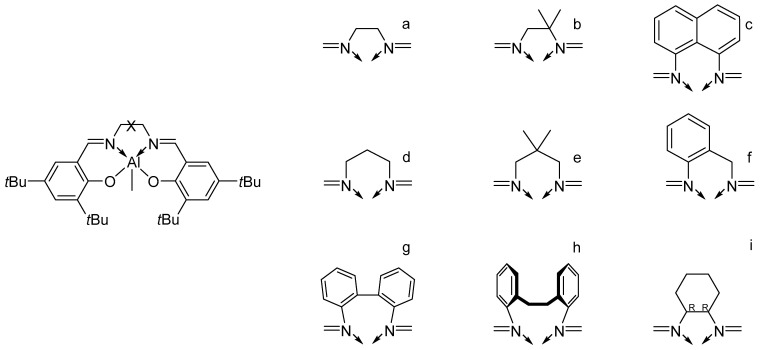
The Al-based catalysts used by Gibson and coworkers (**a**–**h**) [76] and Feijen and coworkers (**i**) [95].

**Figure 21 ijms-25-01647-f021:**
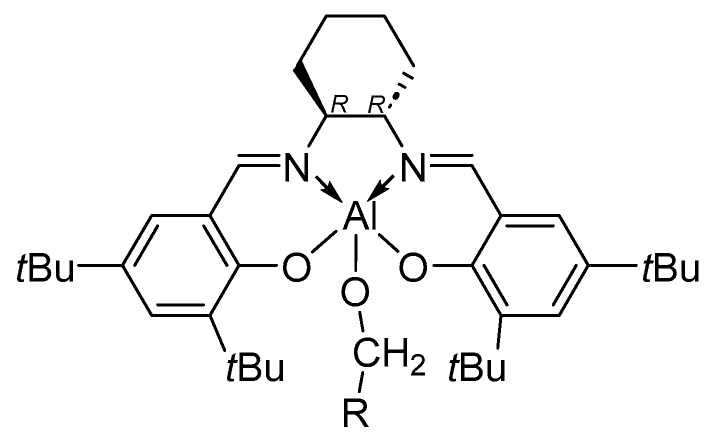
The A-based catalysts used by Chisholm et al. with R = Me, *^i^*Pr, *^t^*Bu, CH(S)MeCl [96].

**Table 1 ijms-25-01647-t001:** Experimental results obtained by Spassky et al. [83].

[LA]:[Cat.]	T [K]	t [h]	Conv. [%]	*M*_n_^exp^·10^−3^ [kg/mol]	*M*_n_^calc^·10^−3^ [kg/mol]	${\left[\alpha \right]}_{D}^{25}$	*OP*
75:1	343	5	19.0	2.7	2.0	137	0.88
75:1	343	3.5	38.0	2.5	4.0	125	0.80
75:1	343	22	62.5	4.8	7.0	106	0.68
75:1	343	42.5	72.0	5.8	8.1	80	0.51
75:1	343	113	90.0	6.8	9.7	20	0.13
75:1	343	281	97.5	5.9	1.1	5	0.30

**Table 2 ijms-25-01647-t002:** Experimental results from phosphasalen M-based catalysts.

Metal	[Cat.]:[*i*PrOH]:[LA]	T [K]	t [h]	Conv. [%]	*M*_n_^exp^·10^−3^ [kg/mol]	*M*_n_^calc^·10^−3^ [kg/mol]	PDI	*P* _i_	Ref.
Y	1:0:500	298	0.5	87	210.0	62.6	1.09	0.76	[83]
	1:1:500	298	1	92	41.0	66.2	1.05	0.74	[83]
	1:1:100	298	0.3	92	12.8	13.2	1.03	0.77	[83]
	1:1:100	298	3	77	16.0	11.1	1.01	0.84	[83]
Lu	1:0:500	298	8	81	101.7	58.3	1.06	0.80	[91]
	1:1:500	298	9	84	38.9	60.5	1.07	0.75	[91]
	1:0.5:500	257	72	84	69.6	60.5	1.02	0.84	[91]
	1:0.5:200	257	48	90	36.0	26.0	1.02	0.83	[91]
La	1:1:500	298	20 s	98	57.3	70.6	1.05	0.28	[91]
	1:2:1000	298	20 s	93	50.0	67.0	1.03	0.28	[91]

**Table 3 ijms-25-01647-t003:** Experimental results obtained by Tang, Chen and coworkers [64,65].

Cat.	[*rac*-DL]:[Cat.]	t [min]	Conv. [%]	*M*_n_^exp^·10^−3^ [kg/mol]	PDI	*I**	*P* _m_	[*mm*] [%]	*T*_m_ [°C]
a	20/1	20	100	4.77	1.17	74	0.91	87	128/136
	50/1	20	100	10.9	1.05	80	0.93	87	133/143
	100/1	20	100	23.0	1.04	75	0.94	89	136/145
	200/1	20	100	32.0	1.03	108	0.93	89	146
b	100/1	20	100	25.1	1.03	69	0.95	89	147
	200/1	20	100	37.3	1.01	93	0.95	88	147
c	100/1	20	100	25.7	1.11	67	0.96	93	153/157
	200/1	20	100	52.7	1.14	66	0.96	94	156
d	100/1	20	100	20.1	1.07	86	0.99	98	161
	200/1	20	100	37.4	1.07	92	>0.99	>99	164
	400/1	20	100	64.3	1.02	107	>0.99	>99	169
	800/1	60	98	119	1.03	113	>0.99	>99	170
	1200/1	30	71	154	1.01	95	>0.99	>99	171

**Table 4 ijms-25-01647-t004:** The experimental results obtained by Gibson and coworkers [76] and Feijen and coworkers [95].

Cat.	[LA]:[Cat.]	T [K]	t [d]	Conv. [%]	*M*_n_^exp^·10^−3^ [kg/mol]	*M*_n_^calc^·10^−3^ [kg/mol]	PDI	*P* _i_	Ref.
a	50:1	343	-	>90	5.9	7.2	1.27	0.83	[76]
b	50:1	343	-	>90	7.2	7.2	1.31	0.77	[76]
c	62:1	343	2	21.1	2.5	1.9	1.04	0.92	[95]
	62:1	343	4	36.3	3.5	3.2	1.04	-	[95]
	62:1	343	24	87.8	8.4	7.8	1.08	-	[95]
	100:1	383	6	94.0	13.9	13.5	1.31	-	[95]
	200:1	403	2	86.4	23.7	24.7	1.18	-	[95]
d	50:1	343	-	>90	9.5	7.2	1.07	0.88	[76]
e	50:1	343	-	>90	8.1	7.2	1.19	0.64	[76]
f	50:1	343	-	>90	7,9	7.2	1.08	0.86	[76]
g	50:1	343	-	>90	6.5	7.2	1.26	0.63	[76]
h	50:1	343	-	>90	8.1	7.2	1.20	0.65	[76]

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
