# Peer review of "Ring Opening Polymerization of Six- and Eight-Membered Racemic Cyclic Esters for Biodegradable Materials"

_ijms, 2024, doi:10.3390/ijms25031647_

Round 1
Reviewer 1 Report
Comments and Suggestions for Authors
The article is of medium extent with appreciable number of well balanced references
In my opinion, the abstract does not characterize the review article content too aptly.
The authors classify polymer materials according to the eco-friendliness of raw materials and ease of their post-use sustainable handling.
They concentrate on two polymers, poly(lactic acid) and poly(3-hydroxybutyrate), which they chose for using of renewable raw materials and possibility of an efficient recycling and degradation process.
They collected from literature information on their properties, using, possible ways of production. In particular, they collected knowledge on a role of stereoselectivity in ring opening polymeration and salen catalysts.
In my opinion, the review article should involve more than two polymers. Despite this, the article may be useful for readers.
I am not sure how much the Conclusion heading fits the real content of this chapter. The link between the contents of previous parts and this chapter seems me not to be straightforward and trivial. Authors can consider, among other, adding references to previous parts of the article (e.g. 4.2, 6.3) to passages with conclusions connected with those parts.
I respect the choice of authors which materials, methods and aspects to include, therefore I recommend to accept with possible minor refinements, despite some reservations expressed above.
In formal aspects, the article is on higher level than an average of articles in MDPI journals. Some tiny formal mistakes (e.g. doubled fullstop after RDS in Conclusions) should be fixed; since the line numbering was removed, I abstain from pointing at specific passages what becomes less ease than in a manuscript with numbered lines.
I appreciate that authors included the list of used abbreviations.
In ref. 8, I am not convinced that it is appropriate to handle "U.N. Secretary General" like with a name and surname.
In ref. 27, odd quotes have remained.
Author Response
Comments and Suggestions for Authors
The article is of medium extent with appreciable number of well balanced references
Our answer: We appreciate the positive comments.
In my opinion, the abstract does not characterize the review article content too aptly.
Our answer: We have rephrased the Abstract, once removed some sections and improved some other ones. We hope that now the message is more clear.
The authors classify polymer materials according to the eco-friendliness of raw materials and ease of their post-use sustainable handling.
They concentrate on two polymers, poly(lactic acid) and poly(3-hydroxybutyrate), which they chose for using of renewable raw materials and possibility of an efficient recycling and degradation process.
They collected from literature information on their properties, using, possible ways of production. In particular, they collected knowledge on a role of stereoselectivity in ring opening polymeration and salen catalysts.
In my opinion, the review article should involve more than two polymers. Despite this, the article may be useful for readers.
Our answer: We have included another polymer, PGLA, and we have made the discussion more general.
I am not sure how much the Conclusion heading fits the real content of this chapter. The link between the contents of previous parts and this chapter seems me not to be straightforward and trivial. Authors can consider, among other, adding references to previous parts of the article (e.g. 4.2, 6.3) to passages with conclusions connected with those parts.
Our answer: We have improved or at least modified those sections, together with the corresponding references. In addition, the conclusions, mainly the first part has been rephrased.
I respect the choice of authors which materials, methods and aspects to include, therefore I recommend to accept with possible minor refinements, despite some reservations expressed above.
In formal aspects, the article is on higher level than an average of articles in MDPI journals. Some tiny formal mistakes (e.g. doubled fullstop after RDS in Conclusions) should be fixed; since the line numbering was removed, I abstain from pointing at specific passages what becomes less ease than in a manuscript with numbered lines.
I appreciate that authors included the list of used abbreviations.
Our answer: Again we appreciate the positive comments.
In ref. 8, I am not convinced that it is appropriate to handle "U.N. Secretary General" like with a name and surname.
In ref. 27, odd quotes have remained.
Our answer: We have corrected those mistakes and other typo errors throughout the manuscript accordingly.
Reviewer 2 Report
Comments and Suggestions for Authors
The focus of this paper could be very interesting. The current version of the paper contains too many topics and is relatively difficult to follow.
1) Please eliminate Sections 1 and 2. This material is not novel and does not contribute to the paper's focus.
2) Section 3 needs major revisions. Please clearly explain the polymerization routes and clarify which routes are used for commercial grades of PLA. Link the polymerization routes, the ratios of isomers, and any critical polymerization conditions to critical properties of the PLA materials. The extensive section on PLA applications should be eliminated because they are not the focus of this paper.
3) Section 4 needs major revisions. Please clearly explain the polymerization routes and clarify which routes are used for commercial grades of P-3HB. Link the polymerization routes, the ratios of isomers, and any critical polymerization conditions to critical properties of the P-3HB materials. The extensive section on P-3HB applications should be eliminated because they are not the focus of this paper.
4) Section 5 needs revisions. Please clearly explain major findings from prior studies.
5) Section 6 needs an introductory sentence or two. Why are you discussing Salen catalysts and what other catalysts are used for ROP, particularly for PLA and 3-PHB?
6) Section 6.1 needs to tie the advantages to other catalysts used for ROP.
7) Section 7 needs some revision. It does not really fit with the rest of the paper and the last paragraph should be eliminated.
8) Abstract and Conclusions - should be revised to clearly focus on ROP.
Comments on the Quality of English LanguageThe paper needs greater focus and clarification of critical points.
Author Response
Comments and Suggestions for Authors
The focus of this paper could be very interesting. The current version of the paper contains too many topics and is relatively difficult to follow.
1) Please eliminate Sections 1 and 2. This material is not novel and does not contribute to the paper's focus.
Our answer: We have reshaped nearly the whole manuscript and even the title. Now the paper is more general, and thus, we removed the word “stereoselective” in the title. If for Sections 1 and 2 we consider the introduction, it is just to settle the topic to more general audience, and get more impact. But if the reviewer still considers that with the changes performed we have to remove those sections, we will do it, simply introducing from Section 3.
2) Section 3 needs major revisions. Please clearly explain the polymerization routes and clarify which routes are used for commercial grades of PLA. Link the polymerization routes, the ratios of isomers, and any critical polymerization conditions to critical properties of the PLA materials. The extensive section on PLA applications should be eliminated because they are not the focus of this paper.
Our answer: We have included a more general discussion. Now all the discussion on applications has gained importance, and we apologize for the past confusing version of the manuscript.
3) Section 4 needs major revisions. Please clearly explain the polymerization routes and clarify which routes are used for commercial grades of P-3HB. Link the polymerization routes, the ratios of isomers, and any critical polymerization conditions to critical properties of the P-3HB materials. The extensive section on P-3HB applications should be eliminated because they are not the focus of this paper.
Our answer: It has been rewritten, specially adding the discussion on PGLA.
4) Section 5 needs revisions. Please clearly explain major findings from prior studies.
Our answer: We have rewritten the section with P3HB comments.
5) Section 6 needs an introductory sentence or two. Why are you discussing Salen catalysts and what other catalysts are used for ROP, particularly for PLA and 3-PHB?
Our answer: We apologize for the confusing text in this section and it was reshaped.
6) Section 6.1 needs to tie the advantages to other catalysts used for ROP
Our answer: We have introduced some pieces of text in this section to remark the advantages.
7) Section 7 needs some revision. It does not really fit with the rest of the paper and the last paragraph should be eliminated.
Our answer: We have done some changes accordingly. And we have removed the discussion on a particular stereoselective system.
8) Abstract and Conclusions - should be revised to clearly focus on ROP.
Our answer: Both sections have been improved.
Comments on the Quality of English Language
The paper needs greater focus and clarification of critical points.
Our answer: We have done our best to improve the manuscript. All changes are highlighted in yellow, including also the parts that have been removed.